# OpenReview forum: "Unbiased Sliced Wasserstein Kernels for High-Quality Audio Captioning"
_NeurIPS.cc/2025/Conference — NeurIPS 2025 poster_

### Official Review · Reviewer_qhDJ · 2025-06-03

**Clarity:** 2
**Significance:** 3
**Originality:** 4
**Rating:** 5
**Confidence:** 2

**Summary:**

This paper introduces a novel kernel-based framework and the Unbiased Sliced Wasserstein RBF (USW-RBF) kernel to enhance cross-modal alignment between audio and language representations in audio captioning tasks. The kernel preserves temporal information using rotary positional embeddings and addresses exposure bias—a common problem in sequence generation models—by facilitating stochastic decoding at inference. The method is presented as a framework called ACUS. Theoretical properties of the kernel (unbiasedness, positive-definiteness) are proven, and extensive empirical evaluations (including humans) across multiple datasets and tasks demonstrate superior performance compared to baselines.

**Questions:**

Some "issues" in "List of detailed issues (excluding typos)" above can be understood as questions, since I am not sure if I understood the reasoning.
For example:
- Why use p-values in Table 3 instead of confidence intervals?
- Do you just add up the loss terms without a weighting hyperparameter?

**Ethical Concerns:**

["NO or VERY MINOR ethics concerns only"]

**Final Justification:**

I have increased my final score since my most important concerns have been resolved, assuming the authors hold their promises.

The resolved concerns were:

- Bad writing and presentation
- lack of overview of the ACUS algorithm
- density assumption in the theoretical contribution

**Limitations:**

Yes. However, I am not familiar with audio experiments, so I might have missed some important limitations which is not mentioned. Further, it would be nice to mention (again) in the limitations that the p.d. property is only proven for densities, while we are using empirical distributions in practice.

**Paper Formatting Concerns:**

No major formatting issues have been noticed.

**Quality:**

2

**Strengths And Weaknesses:**

## Review Summary
First of all, I thank the authors for trying to provide a solid contribution to the scientific community.

Overall, the work presents a novel and clear contribution, offering improved performance on a practically important task.
However, the presentation of the methodology seems too rushed, with a long list of minor issues.
In its current form, I cannot recommend this work to the NeurIPS audience. However, this may very well change with the rebuttal, since, at its core, the contributions are solid.

More details on strengths and weaknesses are presented in the following. Note that I am not familiar with audio tasks, but very familiar with kernel methods; therefore, most of my comments are regarding the presentation of the methodology, and I may miss important shortcomings of the experimental setup.

## Strengths

- The proposed methodology and kernel are novel and original (as far as I know)
- Clear improvement compared to baselines in the experiments
- The methodology neatly includes the positional encodings into the similarity objective without adding much overhead
- Basic theoretical properties of the proposed kernel are proven
- Clear structure of the paper

## Weaknesses

- The quality of the presentation is lacking due to a large quantity of minor issues (c.f. list of detailed issues)
- The language sometimes sounds off, and using any LLMs for writing aid could improve that
- The clarity of the methodology has room for improvement due to various minor reasons, partly also due to the presentation
- The ACUS framework is mentioned as a contribution, but never properly presented in a summarised form (for example, an algorithm box would help)
- Figure 1 has too many unexplained details
- The presented theory would be clearer by sticking more thoroughly to mathematical writing conventions, like defining every used symbol, indicating what a definition and what a statement is, and numbering all equations
- The kernel is only proven to be p.d. for densities, while, in practice, the authors use empirical distributions, which do not have densities (However, I see this more as a minor point since sliced wasserstein kernels are already used in quite a lot of applications and I assume multiple of these also use empirical distributions)

## List of detailed issues (excluding typos)
- "Exposure bias" is explained too late in the paper (I guess it is something very basic in the audio literature, but it would still be nice to have maybe half a sentence for it in the introduction around line 29)
- Line 55: "The proposed kernel is unbiased. Hence, it is highly compatible with stochastic gradient optimization" is a statement the authors make multiple times. At least one citation for this would be nice
- Line 79: Sometimes the input x is bold, sometimes it isn't. More consistency would be nice
- Line 79: The parameter spaces $\Theta$ and $\Phi$ are never used nor described. I guess one can remove them
- Eq. 2: $y_i$ has an index i which has no purpose, is neither defined nor used; further, y is bold but in other instances is not. Again, consistency would be nice
- Line 91: $y_t \in y$ I assume the $y$ should be $Y$, but then you also need to define $Y$
- Line 100: It is clear that $z_x$ is from the encoder model since this was defined earlier. However, $z_y$ and $z_{y^\prime}$ are not clear without being more familiar with the literature
- Line 101: Clarify what $Y$ is. Is it the same as $y$ above?
- Eq. 4: Are the two loss terms added without some weighing hyperparameter $\lambda$? I am not familiar enough with the literature
- Eq. 5: Same issue with $y_i$ as in Eq. 2 plus inconsistency by using bold and non-bold
- Line 123: "Wasserstein distance [14] between µ and ν be two distributions
belongs to Pp(Rd) is defined as" sounds off
- Eq. after line 124: Missing equation number
- Figure 1: Figure does not fit here. Either make it more abstract, so one can understand it without looking up the notation (e.g. UK), or move it further down once the notation is defined. Further, the arrows in the figure might need additional explanations.
- Line 125: The sentence is difficult to read.
- Eq. 6: The symbol for the unit sphere is used but only explained later. Further, the symbols for push-forward measures is used but never defined or explained.
- Line 133: The sentence "As a result, the quantile functions can be approximated efficiently" alone is not very clear. The push-forward projections into an univariate space are used to make the empirical distribution have an inverse (the quantile function), which one can use in practice. An explanation, which makes this more understandable, would be nice.
- Line 134: Is the Monte Carlo estimation proposed in the cited works? I did not see it, and the paragraph does not make it clear if this comes from the authors or if it is picked up from the earlier citations.
- Eq. before line 137: Missing equation number, and the norm has a lower p, while before it hadn't
- Line 137: $\sigma_1$ and $\sigma_2$ might be clearer to be $\sigma_x$ and $\sigma_y$ for a clearer association with x and y. Further, the small $l$ should be dropped in the definition as it is an index for the function output (or at least explain the function in a way such that the index is included)
- Line 138: Sentence sounds off
- Line 140: Maybe do not write "we can define" when the definition comes from the literature, to not confuse the reader
- Line 148: $\kappa$ is no-where used. I assume $\gamma$ maybe?
- Line 153: "Celebrates" sounds off. Also, the representer theorem is not used, so no need to mention.
- Line 162: Missing $p$ in $UK(u,v)$
- Line 165: Why is it crucial? Cite or provide evidence
- Eq. 11: Use \operatorname for $concat$ and $pos$ (also everywhere outside of Eq. 11)
- Eq. 12: What are $\sigma_{\psi, 1}$ and $\sigma_{\psi, 2}$? Further, Eq. 12 mixes a definition with consecutive statements, which is confusing. Further, the requirements of the statements are not clear. Why define the $K$'s? Is this the implementation? If so, clarify why this should be here or move to appendix.
- Eq. 13: Is there no hyperparameter weighting the loss terms?
- Line 194: Drop "the" before "Figure 1". Further, Figure 1 does not demonstrate but only gives an overview.
- Eq. 14: Index $i$ is used without purpose or explanation. Further, $\mathbf{y}$ and $y$ are used. Further, } is missing at the end.
- Line 197: Same issue as in Eq. 14
- Line 216: "we develop a new framework, called ACUS" but the framework is never properly defined. Maybe an algorithm box would help?
- Line 220: "enormous"
- Table 1: "our method" -> "ACUS (ours)"; "ROUGE_L" -> "ROUGE-L"
- Table 3: Previously confidence intervals where used, but here it is p-values. Is there a reason why using p-values for this particular table? Imo confidence intervals are preferable
- Table 2 and 3: You may add "ours" to ACUS
- Line 262: Experiments here do not prove, but demonstrate or suggest
- Table 4: I would prefer confidence intervals and a drop of the "average" column
- Appendix: An overview would be nice.

## Typos
- Line 50: Captioing -> Captioning
- Line 63: "To sum up, our contributions can be summarized" -> "Our contributions can be summarized"
- Line 126: A comma before "i.e." is missing (also at other instances)
- Line 177: Drop -th in $n$-th
- Line 183: "mapping" -> "mappings"
- Line 184: "Let denote the"
- Line 192: Missing space after citation.
- Line 209: ", contrastive loss."
- Line 210: "to combat with"
- Line 213: "is yet successful"
- Line 221: "alginment"
- Line 259: "ROUGE_L" -> "ROUGE-L"
- Line 268: "ROUGE_L" as before
- Line 281: "50 audio"
- Line 295: "but less focus" does not fit the sentence
- Line 311: "CompaA-R"

---

> ### Author Rebuttal · Authors · 2025-07-28
>
> We appreciate the valuable time and feedback of the reviewer to improve your work. We have revised our manuscript and would like to address your concerns outlined below
>
> *We will fix all the typos, revise the sound off sentences, and improve the mathematical symbols based on the reviewer's suggestion to make the manuscript more accessible to readers.*
>
> > **Question 1:** The quality of the presentation is lacking due to a large quantity of minor issues (c.f. list of detailed issues)
> >- The language sometimes sounds off, and using any LLMs for writing aid could improve that
> >- The clarity of the methodology has room for improvement due to various minor reasons, partly also due to the presentation
> >- The ACUS framework is mentioned as a contribution, but never properly presented in a summarised form (for example, an algorithm box would help)
> >- Figure 1 has too many unexplained details
> >- The presented theory would be clearer by sticking more thoroughly to mathematical writing conventions, like defining every used symbol, indicating what a definition and what a statement is, and numbering all equations
>
> We appreciate the detailed feedback of the reviewer for helping us improve the readability of our manuscript. We will revise the manuscript according to reviewer’s feedback. We will add an algorithm box for the ACUS framework in the appendix. We detailed all the necessary information in the caption of Figure. 1, however, we will add more detail for clarity, like the reference to each equation within the Figure. 1 in its caption. We will revise the presentation of our theory and define all the mathematical symbols. We also emphasize that if we overdefine the mathematical symbols, there is not enough space for other content. Therefore, we will carefully revise our theory presentation according to the review’s comments. There are some non-indexed equations, as they represent the decomposition of their parent equation, which are indexed for reference. Thus, indexing might be redundant for them. Finally, we will add a clear indication for definitions and statements for our kernel development section.
>
> > **Question 2:** The kernel is only proven to be p.d. for densities, while, in practice, the authors use empirical distributions, which do not have densities (However, I see this more as a minor point since sliced wasserstein kernels are already used in quite a lot of applications and I assume multiple of these also use empirical distributions)
>
> Thank you for your insightful comments. The condition of having densities is actually redundant since the  Wasserstein kernel is also positive definite for discrete measures  [1] [2]. The difference is only in the way of choosing the embedding function to satisfy Hilbert space properties. Our proof only relies on the positive definite property of the Wasserstein kernel, hence it still holds for distributions with atoms. We will revise the claim accordingly and add the discussion to the revision of the paper.
>
> [1] Sliced Wasserstein Kernel for Persistence Diagrams, Carrière et al.
>
> [2] Distribution Regression with Sliced Wasserstein Kernels, Meunier et al.
>
> > **Question 3:** Why use p-values in Table 3 instead of confidence intervals?
>
> We use the p-value in the Table 3 to demonstrate that the subjective evaluation is statistically significant under the hypothesis test. Regarding hypothesis testing, our null hypothesis is that the difference between the mean values of the subjective evaluation results for the two methods is zero. The null hypothesis is rejected with the $\text{p-values}<0.05$, which indicates that the observed differences in subjective evaluation outcomes are statistically significant rather than attributable to random variation. This statistical validation reinforces the reliability of our human assessment results, while the confidence intervals are not capable of demonstrating the validity of our subjective evaluation.
>
> > **Question 4:** Do you just add up the loss terms without a weighting hyperparameter?
>
> Yes, there is no weighting hyperparmeter in our loss function. We acknowledge that hyperparameter tuning for the weighting parameter could slightly increase the performance of the audio captioning task. However, we do not perform hyperparameter tuning for the weighting parameter in the loss function to demonstrate that our method is not sensitive to the weighting hyperparameter.

---

> > ### Comment · Reviewer_qhDJ · 2025-08-01
> >
> > I have read the author's response, and, assuming the paper is updated as they claim, I am feeling more positive about the paper.
> > The core contributions are solid, and my main concerns were the writing and presentation, the lack of an overview of the ACUS framework, and the density assumption in the theory.
> > I am aware that defining the mathematics takes more space; however, in my opinion, it is more important to offer rigorous statements than to broadly explain statements with unclear variables. The less properly you define your variables, the more explaining you will have to do to not confusing the reader. One way to reduce the space is to carefully think about each equation and really consider if it is necessary to show it in the way you do. Sometimes, some equations are also minor enough to just embed them in the text without giving them their own line.
> > If you consider this in your revision, I am feeling quite positive about your work.
> >
> > In summary, my most important concerns have been resolved, assuming the authors hold their promises, which makes me comfortable recommending this paper to the NeurIPS audience. Good job! :)

---

> ### Author Response · Authors · 2025-08-01
> **Thank you**
>
> We appreciate your time and constructive feedback.
>
> We will carefully revise the manuscript, especially the kernel development section, according to your comments to strengthen the technical rigor of our manuscript.

---

### Official Review · Reviewer_eifA · 2025-06-26

**Clarity:** 2
**Significance:** 2
**Originality:** 2
**Rating:** 5
**Confidence:** 4

**Summary:**

This paper presents a new regularisation technique for mitigating the exposure bias in audio captioning. Unlike the contrastive approach, which doesn’t account for the temporal information in the audio and text due to time aggregation, the paper leverages (an unbiased estimate of) the sliced Wasserstein distance between the sequence of audio and text embeddings. After proving the unbiasedness and the convergence rate of the estimator with Monte-Carlo approximation, quantitative experiments are performed on Audio Captioning and Audio Reasoning, comparing to the MLE baseline, Contrastive approach, and other baselines in the field.

**Questions:**

Did you make an ablation using SW-RBF instead of USW-RBF as kernel for sanity check?

Why does the decoding objective (14) make sense? Are the range of values of the Unbiased Sliced Wasserstein Kernels compared to $p(y_i \mid x)$ compatible for this argmax selection based on the addition of the two terms? For contrastive learning it makes sense because cos takes values in $[-1,1]$.

How do you explain that increasing $L$ doesn’t always monotonically improve the performance in Table 10? Did you try with $L = 1$ as well?

How do you explain the improvement in caption length and lexical diversity in Table 2? Is the increase in caption length and lexical diversity due to the fact that the EOS is less likely to be sampled?

**Ethical Concerns:**

["NO or VERY MINOR ethics concerns only"]

**Final Justification:**

The authors have addressed my concerns during the rebuttal. I agree with Reviewer qhDJ that the main issues with the paper relate to clarity, writing, and notation. Provided the authors follow through on their promised revisions, the paper deserves acceptance.

**Limitations:**

Limitations are assessed in Appendix A.4, showing that the method is more costly than vanilla MLE and MLE + Contrastive Learning. I believe this section should be in the main paper (or at least references to it). The introduction of the bandwidth $\gamma$ as another hyperparameter (for which ablation is conducted in Table 9) should be further acknowledged as a limitation. The hyperparameters should have been selected based on the performance on a held-out validation set (and not the test sets).

**Paper Formatting Concerns:**

Errors / typos:
* L108: $g_{\phi}$ instead of $g_{\theta}$ in (5) ?
* L196: $y_i$ in (14) in UK instead?
* L813: $c_i \in \mathbb{R}^{*}$ (Proof of prop 1)
* The scores of the method are in bold even when another method achieves the same score, e.g., METEOR in Table 6, and Table 9.
* The standard deviations in the RTF computation should be computed in Table 12.

**Quality:**

3

**Strengths And Weaknesses:**

Strengths: * The paper is well motivated, and the method is sound.

Weaknesses: *Graphics quality is bad (e.g., Figure 1). The paper is not very well written.

* The comparison with MLE with a beam size of $5$ is not entirely fair, because your method generates $30$ captions by sampling, therefore multiplying by $\sim6$ the inference cost (in terms of the number of forward passes at inference). To be entirely fair, you should have a higher beam size (30 ideally) for the baseline. Did you use a beam size of $5$ for the RTF inference time measurement in Table 2 ?

* The paper is putting too much emphasis on the fact that the approximation error is diminishing at a parametric rate of $\mathcal{O}(L^{−1/2})$ with $L$ samples. While it is important to mention it, it should not be claimed as a main contribution because this is a well-known result with any unbiased Monte-Carlo estimator with finite second moments.

---

> ### Author Rebuttal · Authors · 2025-07-29
>
> We appreciate the valuable time and feedback of the reviewer to improve your work. We have revised our manuscript and would like to address your concerns outlined below
>
> > **Question 1:** The comparison with MLE with a beam size of 5 is not entirely fair, because your method generates 30 captions by sampling, therefore multiplying by ∼6 the inference cost (in terms of the number of forward passes at inference). To be entirely fair, you should have a higher beam size (30 ideally) for the baseline. Did you use a beam size of 5 for the RTF inference time measurement in Table 2?
>
> The number of generated captions for the MLE baseline using the beam search method is indeed larger than the number of generated captions of our method. Beam search expands each current hypothesis (generated caption) by the beam size at every decoding step. Then, only the top-k (k=beam_size) hypotheses are retained based on their probability scores for the next decoding step until the EOS token is reached. With the beam size of 5, the MLE method generates roughly $M*5$ captions, where $M$ is the sequence length. Given average caption lengths of 7.52 words for AudioCaps and 11.23 words for Clotho test sets (refer to Table 2), the MLE baseline with the beam size of 5 generates about 37.6 and 56.15 captions on the AudioCaps and Clotho test sets, respectively. We conducted the experiments with a beam size of 5 since all the baseline models use this beam size setting in their experiments. Yes, I use the beam size of 5 for the RTF experiments in Table 12 in Appendix A4.
>
> > **Question 2:** The paper is putting too much emphasis on the fact that the approximation error is diminishing at a parametric rate of  O(L−1/2) with L samples. While it is important to mention it, it should not be claimed as a main contribution because this is a well-known result with any unbiased Monte-Carlo estimator with finite second moments.
>
>  Thank you for your insightful comment. We will revise our paper accordingly. Our aim is to highlight the fact that the new sliced Wasserstein kernel can be estimated unbiasedly, hence, we can achieve the standard Monte Carlo error. In addition, by being the expectation of a random one-dimensional Wasserstein kernel, the new sliced Wasserstein kernel might enjoy new sampling tools that can improve the estimation rate. Since we are dealing with the SW kernel instead of the original SW, some investigation should be made to connect the two kinds of literature [1].
>
> [1] A User's Guide to Sampling Strategies for Sliced Optimal Transport, Sisouk et al.
>
> > **Question 3:** Did you make an ablation using SW-RBF instead of USW-RBF as kernel for sanity check?
>
>  | Dataset | Kernel | METEOR | ROUGE_L | CIDEr | SPICE | SPIDEr |
> |---------|--------|--------|---------|-------|-------|--------|
> | AudioCaps | Biased kernel | 0.255 ± 0.002 | 0.495 ± 0.002 | 0.782 ± 0.004 | 0.184 ± 0.003 | 0.485 ± 0.003 |
> | | Unbiased kernel | 0.262 ± 0.001 | 0.509 ± 0.001 | 0.807 ± 0.003 | 0.192 ± 0.001 | 0.5 ± 0.002 |
> | Clotho | Biased kernel | 0.183 ± 0.002 | 0.38 ± 0.001 | 0.401 ± 0.004 | 0.128 ± 0.003 | 0.271 ± 0.004 |
> | | Unbiased kernel | 0.186 ± 0.001 | 0.38 ± 0.001 | 0.419 ± 0.004 | 0.133 ± 0.001 | 0.275 ± 0.003 |
>
> I would like to provide the ablation study using SW-RBF instead of USW-RBF above. As shown in the above table, our unbiased kernel outperforms the biased kernel significantly in five quantitative evaluation metrics, thus, it demonstrates the advantages of the unbiased kernel against the biased kernel in our framework.
>
> > **Question 4:** Why does the decoding objective (14) make sense? Are the range of values of the Unbiased Sliced Wasserstein Kernels compared to p(yi∣x)compatible for this argmax selection based on the addition of the two terms? For contrastive learning it makes sense because cos takes values in [−1,1].
>
> The decoding objective (14) makes sense because both terms of the objective measure the similarity between audio and generated caption, and they are bounded in the range from $0$ to $1$. The Unbiased Sliced Wasserstein Kernels $\mathcal{UK}_\gamma  (Z_x,Z_y;2)=  \mathbb{E}\_{\psi \sim \mathcal{U}(\mathbb{S})\^{d-1}} [ \exp \left( -\gamma W_p^p(\psi \sharp Z_x,\psi\sharp Z_y)\right)  ]$ takes value in $[0,1]$ since the $W_p^p(\cdot)$ is bounded in range from $0$ to $\infty$, and $\gamma > 0$.
>
> > **Question 5:** How do you explain that increasing  L doesn’t always monotonically improve the performance in Table 10? Did you try with L=1 as well?
>
> Thank you for pointing that out. Increasing $L$ can help to reduce the approximation error by reducing the variance of the estimation. In the context of training deep neural networks, having some variance can have a beneficial side effect, which is escaping local optima through the stochastic gradient. That is why we see that the performance might not always be improved when we have a large enough $L$. For $L=1$, we believe that choice is too small given the dimension of the embedding. To consolidate our argument, we would like to provide the experiment with $L=1$ as below
>
> | Dataset | Number of L | METEOR | ROUGE_L | CIDEr | SPICE | SPIDEr |
> |---------|-------------|--------|---------|-------|-------|--------|
> | AudioCaps | L = 1 | 0.257 ± 0.002 | 0. 497± 0.004 | 0. 0.791± 0.008 | 0. 0.189± 0.003 | 0.491 ± 0.005 |
> | | L = 10 | 0.261 ± 0.001 | 0.505 ± 0.002 | 0.793 ± 0.008 | 0.197 ± 0.001 | 0.495 ± 0.005 |
> | | L = 50 | 0.262 ± 0.001 | 0.509 ± 0.001 | 0.807 ± 0.003 | 0.192 ± 0.001 | 0.5 ± 0.002 |
> | | L = 100 | 0.266 ± 0.001 | 0.503 ± 0.002 | 0.805 ± 0.008 | 0.193 ± 0.001 | 0.501 ± 0.003 |
> | Clotho | L = 1 | 0.181± 0.001 | 0.374 ± 0.001 | 0.401 ± 0.01 | 0.131 ± 0.001 | 0.265± 0.007 |
> | | L = 10 | 0.186 ± 0.001 | 0.376 ± 0.001 | 0.401 ± 0.009 | 0.135 ± 0.001 | 0.268 ± 0.005 |
> | | L = 50 | 0.186 ± 0.001 | 0.38 ± 0.001 | 0.419 ± 0.004 | 0.133 ± 0.001 | 0.275 ± 0.003 |
> | | L = 100 | 0.187 ± 0.001 | 0.382 ± 0.001 | 0.42 ± 0.005 | 0.134 ± 0.001 | 0.275 ± 0.004 |
>
> The above table shows that $L=1$ performs worst for our proposed method, which corresponds to the high approximation error for the USW-RBF kernel. Also, the performance variance increases slightly due to the high approximation error for the USW-RBF kernel with a small $L$.
>
> > **Question 6:** How do you explain the improvement in caption length and lexical diversity in Table 2? Is the increase in caption length and lexical diversity due to the fact that the EOS is less likely to be sampled?
>
> The improvement in caption length and lexical diversity is mainly due to the stochastic decoding techniques extensively discussed in [2]. Our framework is inspired by the analysis in [2] which shows that stochastic decoding methods like top-k or top-p samplings methods can mitigate the exposure bias for language generation. Although the stochastic decoding techniques can remedy the exposure bias of a pretrained audio language model, choosing the best aligned caption from sampled captions is challenging. The probability score is inadequate to select the most aligned caption with a given audio because the probability score tends to select the longest caption, which can lead to a hallucination issue. Therefore, we develop the USW-RBF kernel to better measure the cross-modality similarity as a ranking metric for selecting the best-aligned caption generated from stochastic decoding methods. The text-to-audio experiment in Table 2 demonstrates that our kernel not only improves the caption length and lexical diversity but also enhances the semantic alignment between generated captions and ground-truth audio.
>
> [2] Why exposure bias matters: An imitation learning perspective of error accumulation in language generation, K Arora et al.
>
> > **Question 7:** Limitations are assessed in Appendix A.4, showing that the method is more costly than vanilla MLE and MLE + Contrastive Learning. I believe this section should be in the main paper (or at least references to it). The introduction of the bandwidth $\gamma$  as another hyperparameter (for which ablation is conducted in Table 9) should be further acknowledged as a limitation. The hyperparameters should have been selected based on the performance on a held-out validation set (and not the test sets).
>
> We will definitely refer to the limitation in Appendix A.4 in the camera-ready version for a better understanding of our method. We acknowledge that the limitation on bandwidth selection is inevitable since the USW-RBF is a variant of the RBF-kernel. We conducted an ablation study on the bandwidth parameter for the two datasets' test sets to examine the sensitivity of our method to the value of the bandwidth parameter $\gamma$. The ablation study serves to understand method behaviors and provide insights into parameter influence rather than hyperparameter selection.

---

> > ### Comment · Reviewer_eifA · 2025-08-01
> > **Thank you for your response**
> >
> > I appreciate the response from the authors. The method is well motivated and competitive and the paper deserves acceptance, provided that:
> >
> > * the authors include the rebuttal updates for the final version.
> > * the graphics quality is improved in Figure 1, 2 (you can use, e.g., LaTeX with matplotlib for high-quality rendering)
> > * the issues and typos are corrected (see *Paper Formatting Concerns* and also the review from Reviewer qhDJ).
> >
> > Other minor typos/style issues noticed:
> > * Use \mathrm{} when using letters in the equations, e.g., $\\mathcal{L}\_{\\mathrm{MLE}}$ instead of $\\mathcal{L}_{MLE}$, and use $\mathrm{d}x$ instead of $dx$ in the integrals.
> > * Use consistent font in the Tables (some rows have $$ font while others do not). Also adjust the spacing in the Tables (the first and last \midrule seem misplaced).
> > * The closing brace } is missing in (14)
> > * Including commas or periods after the equations where it is missing

---

> > > ### Author Response · Authors · 2025-08-01
> > > **Thank you**
> > >
> > > We appreciate your time and constructive feedback.
> > >
> > > We will include the rebuttal experiments and discussion in the final version to strengthen the technical rigour of our manuscript. We will also improve the figure quality and fix all writing issues.

---

### Official Review · Reviewer_ceXp · 2025-07-01

**Clarity:** 4
**Significance:** 3
**Originality:** 3
**Rating:** 5
**Confidence:** 3

**Summary:**

This paper presents the challenge of exposure bias in audio captioning models caused by the mismatch between training (mostly due to the teacher-forcing) and inference phases. To overcome the limitations of contrastive learning, particularly its inability to model temporal alignment between audio and text, the authors propose the Unbiased Sliced Wasserstein RBF (USW-RBF) kernel, which incorporates rotary positional embeddings to preserve temporal information. They introduce the ACUS framework, integrating the kernel into training and leveraging it at inference for caption re-ranking using stochastic decoding. Experimental results on AudioCaps and Clotho datasets, as well as audio reasoning benchmarks (CompA-R and MMAU), demonstrate improved caption quality, lexical diversity, and generalization across tasks.

**Questions:**

- How does the model handle cases where the number of tokens in audio and caption representations differs? Is dynamic alignment applied?
- Why was rotary positional embedding favored over absolute or learned embeddings, as it is explicitly mentioned in the paper? Is there an empirical comparison?
- How sensitive is the performance to the number of Monte Carlo samples (L) used in kernel estimation?
- Some additional ablation could be beneficial:
   - Although unbiasedness from the introduction of USW-RBF kernel is theoretically desirable, the variance of the estimator could still be high (depending on \gamma, p, and distributional overlap). It would be better to empirically or theoretically explore variance-bias trade-offs, which are important in kernel-based learning stability.
   - It seems that no regularization or scaling factor is applied when the log-probability and the kernel similarity are combined. Their relative magnitudes may vary, so \gamma may serve a dual role here. This makes it further more important, so it could be good to include some discussion on the selection of this value.

**Ethical Concerns:**

["NO or VERY MINOR ethics concerns only"]

**Final Justification:**

As indicated in previous discussion, I would like to keep my original score.

**Limitations:**

- There is a constraint mentioned in the context (N = M). This restricts flexibility, especially for naturally varying sequence lengths in real-world data. More clarification on this would be beneficial.
- The uniform random sampling of projection directions might not be optimal. More efficient or learned projections could improve estimation quality.
- Human evaluation shows captions are more fluent than human references, possibly due to over-optimization on linguistic smoothness at the cost of factual grounding in audio.

**Quality:**

4

**Strengths And Weaknesses:**

Strengths:
- The paper introduces an unbiased kernel version of the sliced Wasserstein distance specifically for multimodal alignment with temporal consideration, which is originally proposed.
- Comprehensive evaluations show consistent improvements over strong baselines (e.g., Enclap, contrastive learning), particularly on AudioCaps.
- The mathematical formulation of the proposed method is solid.
- The kernel also improves performance on audio reasoning tasks, suggesting utility beyond captioning. This indicates a potential in generalization of the proposed method
- The paper is generally easy to follow, and the reviewer enjoys reading it.


Weakness:
- The paper assumes equal-length latent sequences for audio and caption representations, which is a simplification that may not hold in real-world scenarios. Most sequence‑matching techniques (e.g., DTW, soft‑DTW) inherently support different lengths, but the USW‑RBF method does not clearly address this discrepancy
- On a more diverse Clotho dataset, improvements are less compared to AudioCaps. Maybe due to the complexity of the dataset, but more analysis and reasoning would be beneficial.
- It would be better to provide some error analysis on where/why the method may fail in different scenarios.

---

> ### Author Rebuttal · Authors · 2025-07-29
>
> We appreciate the valuable time and feedback of the reviewer to improve your work. We have revised our manuscript and would like to address your concerns outlined below
>
> > **Question 1:** The paper assumes equal-length latent sequences for audio and caption representations, which is a simplification that may not hold in real-world scenarios. Most sequence‑matching techniques (e.g., DTW, soft‑DTW) inherently support different lengths, but the USW‑RBF method does not clearly address this discrepancy. Also, There is a constraint mentioned in the context (N = M). This restricts flexibility, especially for naturally varying sequence lengths in real-world data. More clarification on this would be beneficial.
>
> Thank you for your insightful comments. The reason we assume $N=M$ is just for having a nice form of SW with one-to-one mapping for projections for presentation. We would like to recall that USW‑RBF can still be computed for $N\neq M$ since SW can be computed in that case. In particular, we can use the north-west corner algorithm to compute one-dimensional transportation maps for SW [1]. We will revise the paper accordingly and add the discussion to the revision.
>
> [1] Computational Optimal Transport,  Peyré et al.
>
> > **Question 2:** On a more diverse Clotho dataset, improvements are less compared to AudioCaps. Maybe due to the complexity of the dataset, but more analysis and reasoning would be beneficial. It would be better to provide some error analysis on where/why the method may fail in different scenarios.
>
> We will add more qualitative samples to the appendix to perform error analysis and delve deeper into the failure cases of our method in the AudioCaps and Clotho datasets, thereby understanding the model's behaviors and providing a discussion on error analysis in the appendix.
>
> > **Question 3:**  How does the model handle cases where the number of tokens in audio and caption representations differs? Is dynamic alignment applied?
>
> As answered in Question 1, the model can also handle the case when the number of tokens in audio and caption representations differs in nature. We will add the new discussion to the revised version.
>
> > **Question 4:** Why was rotary positional embedding favored over absolute or learned embeddings, as it is explicitly mentioned in the paper? Is there an empirical comparison?
>
> We conducted an ablation study in Table 7 in Appendix A.3 to study the effectiveness of different positional embedding techniques. According to the experimental results, the rotary positional embedding outperformed the traditional positional embedding technique, therefore, we chose the rotary positional embedding method to encode the temporal information for acoustic and linguistic features.
>
> > **Question 5:** How sensitive is the performance to the number of Monte Carlo samples (L) used in kernel estimation?
>
> We conducted an ablation study in Table 10 of Appendix A.3 to investigate the sensitivity of our proposed method to the number of Monte Carlo samples, $L$. The experimental results demonstrate that the optimal value for the number of projections, $L = 50$, balances performance and inference time. A lower number of projections can lead to high approximation errors, which decrease model performance. In contrast, a higher number of projections shows a marginal performance gain at the cost of increased inference time.
>
> > **Question 6:** Some additional ablation could be beneficial:
> > - Although unbiasedness from the introduction of USW-RBF kernel is theoretically desirable, the variance of the estimator could still be high (depending on \gamma, p, and distributional overlap). It would be better to empirically or theoretically explore variance-bias trade-offs, which are important in kernel-based learning stability.
> > - It seems that no regularization or scaling factor is applied when the log-probability and the kernel similarity are combined. Their relative magnitudes may vary, so \gamma may serve a dual role here. This makes it further more important, so it could be good to include some discussion on the selection of this value.
>
> We will add an additional analysis to study the variance-bias tradeoff of our proposed method on the test sets of two datasets in the appendix. We conducted the ablation study on the sensitivity of our method in Table 9 and discussed the sensitivity of our method regarding the bandwidth factor $\gamma$ in the Appendix A.3.
>
> > **Question 7:** The uniform random sampling of projection directions might not be optimal. More efficient or learned projections could improve estimation quality.
>
> Thank you very much for your comments. We agree that a non-uniform slicing distribution could lead to better estimation quality. However, we believe that slicing distribution selection for the sliced Wasserstein kernel is a new area which is worth careful investigation. A direct application of recent techniques for SW might not be directly applicable [2].
>
> [2] Energy-Based Sliced Wasserstein Distance, Nguyen et al.
>
> > **Question 8:** Human evaluation shows captions are more fluent than human references, possibly due to over-optimization on linguistic smoothness at the cost of factual grounding in audio.
>
> We acknowledge the limitation pointed out by the reviewer.  The main reason for the oversmoothness of generated captions is due to the decoder of the audio captioning model. The decoder model leverages a language model extensively pretrained on language tasks, and then it is finetuned on the audio-caption pairs for audio captioning. Therefore, the audio captioning model possesses the bias from the text training data of the decoder component. We believe that if we have enough audio-text training data to train the whole audio language models from scratch, the oversmoothness issue could be mitigated.

---

> > ### Comment · Reviewer_ceXp · 2025-07-31
> > **Reply to Rebuttal**
> >
> > I would like to thank the authors' clarification. All my questions and concerns have been addressed. Would like to keep the positive scores from my end.

---

> ### Author Response · Authors · 2025-08-01
> **Thank you**
>
> We are glad our rebuttal addressed your concerns and appreciate that you maintain a positive score.
>
> Please feel free to let us know if you have any further concerns.

---

### Official Review · Reviewer_tM5J · 2025-07-02

**Clarity:** 2
**Significance:** 2
**Originality:** 2
**Rating:** 4
**Confidence:** 3

**Summary:**

This paper targets the exposure bias problem in audio captioning, where models are primarily trained via maximum likelihood estimation without considering negative or contrastive samples. To address this, the authors propose the ACUS framework, which leverages a Wasserstein RBF kernel to model similarities, with reasonable convergence under the unbrella of stochastic gradient optimizations. The main reason to design a new distance metric is that traditional techniques like DTW could suffer from the curse of dimensionality. The method is evaluated on Clotho and AudioCaps, showing improved performance.

**Questions:**

Please see the weakness section.

**Ethical Concerns:**

["NO or VERY MINOR ethics concerns only"]

**Final Justification:**

In the rebuttal, the authors clearly justified the advantage of the USW-RBF distance over prior distance metrics and well explained the rationale behind the chosen baselines. Also, the authors provide additional quantitative and qualitative experiments as well. Therefore, the reviewer increased the score to borderline accept.

**Limitations:**

yes.

**Paper Formatting Concerns:**

No.

**Quality:**

2

**Strengths And Weaknesses:**

Strength:
1. Addressing exposure bias in generative captioning model makes sense; it can be an important problem.
2. The proposed method is technically sound and shows some improvements on standard benchmarks.

Weakness:

1. Motivation: The argument that contrastive learning is less effective for audio than text due to modality differences feels unconvincing (line 34-40). Both audio and text are sequential in nature, and while audio may pose practical challenges (e.g., dimensionality, alignment), the paper should avoid overclaimed. Similarly, while DTW may be suboptimal in high-dimensional settings, it’s unclear why other established distance metrics in the audio domain (e.g., CLAP, Wav2Vec, or HuBERT embeddings) are not considered. Since the motivation for designing the USW-RBF kernel is all based on these limitations, a stronger justification or empirical comparison is encouraged.

2. Evaluation concerns:
	- Missing baselines: The paper misses comparison with several recent and relevant approaches [1–4], which should at least be discussed.
	- Metric sensitivity: Given that most performance differences are marginal across methods, relying solely on old-school captioning metrics may be insufficient. The authors should consider using newer LLM-based evaluation tools [5,6] to assess caption quality.
	- Qualitative analysis: The examples in Appendix A.5 do not show clear improvements over baselines, which weakens the qualitative support for the proposed method.

- [1] CLAP-ART: Automated Audio Captioning with Semantic-rich Audio Representation Tokenizer (Interspeech2025)
- [2] LAVCap: LLM-based Audio-Visual Captioning using Optimal Transport (ICASSP 2025)
- [3] Enhancing Automated Audio Captioning via Large Language Models with Optimized Audio Encoding (Interspeech 2024)
- [4] Taming Data and Transformers for Audio Generation (arXiv 2024)
- [5] CLAIR-A: Leveraging Large Language Models to Judge Audio Captions (arXiv 2024)
- [6] MACE: Leveraging Audio for Evaluating Audio Captioning Systems (arXiv 2024)

---

> ### Author Rebuttal · Authors · 2025-07-28
>
> We appreciate the valuable time and feedback of the reviewer to improve your work. We have revised our manuscript and would like to address your concerns outlined below
>
> > **Question 1:** Motivation: The argument that contrastive learning is less effective for audio than text due to modality differences feels unconvincing (line 34-40). Both audio and text are sequential in nature, and while audio may pose practical challenges (e.g., dimensionality, alignment), the paper should avoid overclaimed. Similarly, while DTW may be suboptimal in high-dimensional settings, it’s unclear why other established distance metrics in the audio domain (e.g., CLAP, Wav2Vec, or HuBERT embeddings) are not considered. Since the motivation for designing the USW-RBF kernel is all based on these limitations, a stronger justification or empirical comparison is encouraged.
>
> We would like to provide the following rationale to explain why we exclude specific established distance metrics, such as Wav2Vec and HUBERT, from our evaluation. Both Wav2Vec and HUBERT embedding models are specifically pretrained for speech representation learning, emphasizing linguistic content comprehension rather than acoustic feature understanding required for audio captioning applications. Consequently, they are suboptimal for quantifying audio-text semantic discrepancies in our context. Additionally, the CLAP metric is cosine similarity pretrained using the contrastive learning technique, making it functionally equivalent to our contrastive learning baseline method described in this work.
>
> We discuss the limitations of the contrastive learning method (CLAP distance metric) in Section 2.2 and conduct both quantitative and qualitative experiments to justify the effectiveness of the USW-RBF distance metric compared with the baseline metric. Mainly, we conducted a subjective evaluation, which is more reliable than LLM-as-a-judge metrics [5,6]. Furthermore, we study the effectiveness of the USW-RBF method and the contrastive learning method for the audio reasoning task in Section 5.3, which requires sophisticated comprehension capabilities from audio-language models. The USW-RBF outweighs the contrastive learning method significantly in the audio reasoning tasks, thus, the USW-RBF is capable of measuring the cross-modality similarity more effectively than the contrastive learning method . We also performed ablation experiments to evaluate various similarity metrics for the audio captioning task, comparing DTW, soft-DTW, and Wasserstein distance approaches (detailed in Table 6, Appendix A.3). Our findings indicate that the proposed USW-RBF metric consistently enhances caption generation performance, demonstrating superior capability in quantifying audio-text semantic alignment compared to baseline methods.
>
> > **Question 2:** Evaluation concerns: Missing baselines: The paper misses comparison with several recent and relevant approaches [1–4], which should at least be discussed.
>
> We appreciate the reviewer for pointing out some potential baselines. We will definitely discuss those baselines in our camera-ready. We also clarify the reason why we did not include the prior works [1,2,3] in our initial draft. The CLAP-ART[1] is a contemporaneous work with our work, the CLAP-ART will be published at Interspeech 2025 in August, while this work was submitted in May. Therefore, it is not possible to discuss the CLAP-ART in the initial draft. The LAVCap [2] is an Audio-visual captioning method, which leverages both visual and acoustic features to generate audio captions, thus, it is incompatible with our work, which generates audio captions from the mere acoustic features.  The prior work in [3] pretrains their AAC model on WavCaps, AudioCaps, and Clotho training data, while we only pretrain our method on either AudioCaps or Clotho training data. Therefore, it is not a fair comparison with our method due to the mismatch in the pretraining data. We compared our method with the AutoCap method [4] in the Table. 1.
>
> > **Question 3:** Metric sensitivity: Given that most performance differences are marginal across methods, relying solely on old-school captioning metrics may be insufficient. The authors should consider using newer LLM-based evaluation tools [5,6] to assess caption quality.
>
> We would like to emphasize that we conducted a subjective evaluation for the audio captioning task in Table.3, which is more reliable than LLM-as-a-judge evaluation methods [5, 6]. The subjective evaluation is conducted with a high inter-annotation agreement to demonstrate the reliability of the experiment. The metrics [5, 6] are not widely-used by the audio captioning community and have not been published in any conferences. Furthermore, recent relevant studies [1-4] do not employ these evaluation measures, raising concerns about their applicability and validity for audio captioning tasks. However, we still conduct supplementary experiments using these two automated metrics [5, 6] for a comprehensive evaluation in the analysis below.
>
> | Dataset | Method | CLAIR-A | MACE |
> |---------|--------|---------|------|
> | AudioCaps | Enclap | 0.686 | 0.597 |
> | | Enclap-CL | 0.698±0.02 | 0.615±0.04 |
> | | ACUS | **0.723±0.03** | **0.634±0.02** |
> | Clotho | Enclap | 0.639 | 0.571 |
> | | Enclap-CL | 0.645±0.02 | 0.578±0.02 |
> | | ACUS | **0.657±0.02** | **0.589±0.02** |
>
> The above results demonstrate the same findings as our subjective evaluation: the ACUS framework outperforms the contrastive learning baseline in generating high-quality audio captions. We will add the additional evaluation in the revised version to better demonstrate the improvement of our proposed method.
>
> > **Question 4:** Qualitative analysis: The examples in Appendix A.5 do not show clear improvements over baselines, which weakens the qualitative support for the proposed method.
>
> We would like to provide more qualitative examples that demonstrate a clear improvement over baselines.
>
> ---
> (More complete environment context)
>
> **Enclap:** A helicopter flying.
>
> **Enclap with CL:** A helicopter flying.
>
> **Enclap with ACUS:** A helicopter flying as wind blows heavily into a microphone.
>
> **Reference:**
> - A helicopter flying followed by wind heavily blowing into a microphone.
> - A motorboat engine revving.
> - A helicopter flying followed by wind heavily blowing into a microphone.
> - An engine running and wind blowing hard.
> - An aircraft motor is operating with rhythmic whirring, then wind roars.
> ---
> (Better sequential event detection)
>
> **Enclap:**  Rain falls then thunder rumbles.
>
> **Enclap with CL:** A man is speaking and rain is falling.
>
> **Enclap with ACUS:** Rain falls and thunder roars in the distance as a man speaks.
>
> **Reference:**
> - It is raining and thundering, and then a man speaks.
> - A man talking as rain falls and thunder roars in the background
> - Distant thunder with rain falling followed by a man speaking loudly
> - Thunder crashes and rain splashes, during which two adult males speak
> - Thunder roaring in the distance as rain lightly pours and a man yells followed by another man humming.
> ---
> (Improve sound recognition)
>
> **Enclap:** A man speaks and a door opens.
>
> **Enclap with CL:** A man talks while some objects are tapped.
>
> **Enclap with ACUS:** Birds are chirping and a man is speaking.
>
> **Reference:**
> - A short hammering sound followed by two men speaking.
> - A man is speaking as birds are chirping.
> - Male speaking and birds chirping.
> - Males speaking and birds chirping.
> - Hammering and then a man speaks followed by rubbing and then a second man speaks.
> ---
> (Improve sound recognition)
>
> **Enclap:** Rain falls and wind blows..
>
> **Enclap with CL:** Wind blows and leaves rustle.
>
> **Enclap with ACUS:** Rain falls and thunder roars in the distance.
>
> **Reference:**
> - Rain falling as birds are chirping followed by thunder.
> - Wind with rain followed by thunder.
> - Rain pitter-patters as thunder cracks and the wind blows.
> - Bird wings flapping then a bird chirping while rain falls followed by thunder.
> - Rain and thunder in a storm.

---

> ### Author Response · Authors · 2025-08-02
> **Thank you**
>
> We are glad our rebuttal addressed your concerns and appreciate that you have increased the score. We will revise our manuscript accordingly and include the rebuttal discussion in the final version.
>
> Please feel free to let us know if you have any further concerns.

---

### Note · Authors · 2025-08-12

Dear Area Chair and Reviewers,

We appreciate your valuable time and feedback to improve your work. We are glad that our rebuttal addressed all concerns from reviewers. We would like to summarize the discussion as follows:

**Novelty:** all reviewers acknowledge the originality and novelty of our work. This work is the first work that proposes a new alignment metric, the Unbiased Sliced Wasserstein RBF (USW-RBF), leveraging the sliced Wasserstein technique for better measuring acoustic and linguistic similarity scores. Based on the new alignment metric, we propose the ACUS framework to address the exposure bias issue for the audio captioning task. Our experimental results demonstrate that the USW-RBF is capable of improving not only the quality of audio caption but also reasoning abilities of audio language models.

**Reviewer's suggestion:**

- Reviewer tM5J: We will incorporate the rebuttal discussion, the additional quantitative experiment, and new qualitative examples in the appendix to strengthen our paper.

- Reviewer ceXp: We will add an error analysis discussion in the final version to better understand our proposed method.

- Reviewer eifA: We will incorporate the rebuttal discussion and resolve all presentation issues, such as the Figure 1 quality and typos.

- Reviewer qhDJ: We will fix all typos and improve mathematical symbols to make the manuscript more accessible to readers

We sincerely thank all reviewers again for their valuable suggestions, which have greatly helped strengthen our paper.

---

### Decision · Program_Chairs · 2025-09-17

**Decision:**

Accept (poster)

**Comment:**

The paper targets the exposure bias of a teaching-forcing autoregressive language model for the audio captioning task. Conventional contrastive-based solutions do not capture the temporal correspondence between audio and caption text. The proposed ACUS method addresses this issue by measuring distance with temporal alignment, where results are demonstrated on multiple benchmarks.

As commented by multiple reviewers, the proposed method is novel and mathematically sound, with strong empirical evidence. On the other hand, the method only tackles a specific task of language models, where a higher rating would come with the expectation of a broader impact. The final version should reflect the suggestions from reviewers and the promises in the rebuttal.